# Glycol Chitosan-Docosahexaenoic Acid Liposomes for Drug Delivery: Synergistic Effect of Doxorubicin-Rapamycin in Drug-Resistant Breast Cancer

**DOI:** 10.3390/md17100581

**Published:** 2019-10-12

**Authors:** Min Woo Kim, Takuro Niidome, Ruda Lee

**Affiliations:** 1International Research Organization for Advanced Science and Technology (IROAST), Kumamoto University, Kumamoto 860-8555, Japan; minwoo-kim@kumamoto-u.ac.jp; 2Faculty of Advanced Science and Technology, Kumamoto University, Kumamoto 860-8555, Japan; niidome@kumamoto-u.ac.jp

**Keywords:** marine pharmaceuticals, glycol chitosan, nanoliposome, combinatorial therapy, drug resistance, drug delivery, cancer therapy

## Abstract

Marine ecosystems are the most prevalent ecosystems on the planet, providing a diversity of living organisms and resources. The development of nanotechnology may provide solutions for utilizing these thousands of potential compounds as marine pharmaceuticals. Here, we designed a liposomal glycol chitosan formulation to load both doxorubicin (DOX) and rapamycin (RAPA), and then evaluated its therapeutic potential in a prepared drug-resistant cell model. We explored the stability of the drug delivery system by changing the physiological conditions and characterized its physicochemical properties. The electrostatic complexation between DOX-glycol chitosan and docosahexaenoic acid RAPA-liposomes (GC-DOX/RAPA ω-liposomes) was precisely regulated, resulting in particle size of 131.3 nm and zeta potential of −14.5 mV. The well-characterized structure of GC-DOX/RAPA ω-liposomes led to high loading efficiencies of 4.1% for DOX and 6.2% for RAPA. Also, GC-DOX/RAPA ω-liposomes exhibited high colloidal stability under physiological conditions and synergistic anti-cancer effects on DOX-resistant MDA-MB-231 cells, while showing pH-sensitive drug release behavior. Our results provided a viable example of marine pharmaceuticals with therapeutic potential for treating drug-resistant tumors using an efficient and safe drug delivery system.

## 1. Introduction

Marine-sourced materials are attracting significant attention because many marine-derived materials are biomedically relevant compounds for cancer treatment, as well as drug delivery systems themselves [1]. Recently, marine pharmaceuticals such as omega-3 fatty acids, chitosan, glucosamine, and fucoidan have been highlighted in research [2,3,4,5]. Exploring new resources can provide innovative drug formulation alternatives to conventional medicines. Out of these alternative options, glycol chitosan (GC) nanoparticles have attracted greater attention due to their strong potential in application as drug carriers; some of GC’s advantages include being able to improve drug solubility, biocompatibility, biodegradability, and easy modification [6,7]. Docosahexaenoic acid (DHA) also has been reported to be beneficial in cancer therapy as a novel liposomal formulation [8,9]. In order to enhance the benefits of potential outcomes and develop novel drug delivery systems in combination with existing drug formulations, new preparation and optimization methods are required.

In conventional cancer therapy, doxorubicin (DOX) is one of the most commonly used anti-cancer drugs and is also effective as a neoadjuvant treatment for breast cancer [10]. Patients with early-stage breast cancer may benefit from DOX-based chemotherapy, but some show recurrent or serious side effects, particularly cardiac toxicity [11]. The primary reasons for the failure of chemotherapy are the absence of effective drug carriers and the acquisition of multi-drug resistant (MDR) cells during treatment. Therefore, treatment with liposomal doxorubicin formulations such as Doxil/Caelyx, LipoDox, and Myocet has been proposed to maximize the payload delivered to the target site [12]. In addition, combination with other chemotherapeutics for anti-MDR has been recommended to reduce the required dose of DOX and the side effects [13,14,15].

The mechanisms that contribute to drug resistance are diverse [16,17]. Therefore, better understanding of the potential pathways for blocking the MDR mechanism will help to enhance current treatments. It has already been well reported that repeated drug treatment can increase the expression of efflux pump like p-glycoprotein, allowing cancer cells to acquire a mechanism to protect themselves against anti-cancer drugs [18]. Another important mechanism is the activation of the anti-apoptotic signal to minimize DNA damage from doxorubicin, which interrupts the proliferation of the cells. Many studies have reported that PI3K-Akt-mTOR signaling, which is involved in cell survival, is activated by continuous chemotherapy [19]. Regulating this pathway could therefore be a key approach to maximizing the therapeutic effect of anti-cancer drugs in MDR cancer. Rapamycin (RAPA) is a specific inhibitor of mTOR signaling and has been shown to be effective against various types of tumor, particularly when combined with DOX [20]. Indeed, the inhibition of mTOR signaling by RAPA may restore sensitivity to chemotherapy in MDR cancer.

In this study, we synthesized and characterized GC-DHA liposomes containing DOX and RAPA. The chitosan polysaccharide and omega-3 fatty acid nanoparticles are considered to be good drug carriers because of their biocompatibility and biodegradability [6,8]. It is challenging to load DOX and RAPA into one particle because of their different physicochemical properties, therefore, we optimized the marine nano-formulation to enable it to carry sufficient anti-cancer chemotherapeutics to treat cancer. We report the physicochemical characteristics of the designed drug delivery system and its potential utility as a drug carrier. In addition, we verified the synergistic effects of DOX and RAPA in DOX-resistant breast cancer cells.

## 2. Results and Discussion

### 2.1. Characterization of GC-DOX/RAPA ω-Liposomes

This study aimed to provide a practical example of a glycol chitosan and liposome complex containing anti-cancer drugs. The conceptual scheme for GC-DOX/RAPA ω-liposomes is illustrated in Figure 1. Hydrophobic RAPA and DHA were co-encapsulated in the lipid bilayer, forming DHA liposomes containing RAPA (RAPA ω-liposomes). A complex was then formed between RAPA ω-liposomes and doxorubicin-conjugated glycol chitosan (GC-DOX) with a pH-sensitive doxorubicin release profile by stirring them together; referred to here as GC-DOX/RAPA ω-liposomes.

The size and zeta-potential of all GC-liposomal formulation ratios were measured immediately after preparation (Figure 2a). RAPA ω-liposomes were 91.61 ± 1.08 nm in diameter with −31.97 ± 2.61 mV surface charge, whereas the particle size when complexed at a 20:1 ratio of liposomes to GC slightly increased to 131.30 ± 2.42 nm in diameter and the surface charge was neutralized from −31.97 ± 2.61 mV to −14.53 ± 0.15 mV. At liposome to GC ratios of 15:1 and higher, the particle size, surface charge, and polydispersity index (PDI) increased significantly. The particles at a 5:1 ratio precipitated and were removed after centrifugation, even though the particle size appeared to slightly decrease to 490.27 ± 62.6 nm compared to that at a 10:1 ratio (Appendix A). The PDI value for particles with a 20:1 ratio decreased from 0.213 to 0.159 compared with bare liposomes. These results can be explained by the electrostatic interaction between the liposomes, which have a strong negative surface charge, and glycol chitosan that has oppositely charged surface amine groups [21] (Figure 1). The size of the particles increased exponentially at a higher GC rate, which indicates that the degree of electrostatic interaction between the liposomes and GC must be well controlled to maintain the stability of the particles. GC-DOX/RAPA ω-liposomes complexed at a 20:1 ratio were found to be optimal for use in subsequent experiments and showed unimodal size distribution with no tendency to aggregate (Figure 2b). 

Morphological studies on the shape of the particles using transmission electron microscopy (TEM) also revealed that RAPA ω-liposomes were spherical with a diameter below 100 nm. Notably, as GC-DOX was added to the RAPA ω-liposomes, liposomal complexes could be observed; generally, the clustering of liposomes is termed as aggregation, however, we refer to it as complexation because the particles maintained a small size. It is thought that linear GCs can act as an adhesive between the liposomes and fill the void space to make a more condensed structure (Figure 2c). Although previous reports have described a form of liposomes coated with chitosan [22,23,24], our experimental data showed different characteristics. There is no clear explanation for our observations, however, they may be due to the characteristics of GC. The GC used in this study has a relatively low molecular weight (~82,000) compared with that used in previous reports that have shown GC coating patterns (~400,000). The length of chitosan molecular chains determines the solubility, formation of aggregates, hydrogen bonding, and hydrophobic interactions [25,26]. It was advantageous to conjugate low molecular weight GC to DOX, as it possesses better solubility and stability at physiological pH [27]. Moreover, since the amine functional groups of chitosan are consumed in the reaction with DOX, the properties of the conjugate could be different to those of non-functionalized GC.

We concluded that adding GC affects the shape of the particles, but not their stability. GC-DOX/RAPA ω-liposomes maintained a low PDI and showed uniform shape when they were observed by TEM. In addition, hydrophobic RAPA encapsulated in the liposomes and DOX conjugated with GC showed good encapsulation efficiency (EE) and loading efficiency (LE) at the ratio that resulted in the optimal size. The EE and LE of RAPA were 69.9 ± 2.3% and 6.2 ± 0.2%, and the EE and LE of DOX were 90.3 ± 4.7% and 4.05 ± 0.2%, respectively (Figure 2d).

### 2.2. Influence of pH on Stability of GC-DOX/RAPA ω-Liposomes

Since pH decreases are observed in the tumor microenvironment as well as in the endocytosis process, the design and development of pH-sensitive drug delivery systems plays a crucial role in efficient cancer therapy [28]. GC-DOX/RAPA ω-liposomes are designed to become unstable in response to decreases in pH. To evaluate their pH stability, four different pH phosphate buffered saline (PBS) solutions were prepared and GC-DOX/RAPA ω-liposomes were incubated at various pH values (pH 4.0, pH 5.0, pH 6.5, or pH 7.4). The distribution of particle size and zeta-potential were then observed after 1 day. As shown in Figure 3a, the particle size was stable near pH 7.4 and pH 6.5, however, we observed aggregation of the particles from pH 5. The average size of the particles also continued to increase significantly when they were incubated in pH 5 and pH 4 for a long time. This can be explained by the change of surface electrical charge in response to pH stimuli. The surface charges of the particles at pH 7.4 and pH 6.5 were −14.5 ± 0.2 mV and −11.3 ± 0.8 mV, respectively. In pH 5.5 and pH 4 solutions, the surface charge increased to 3.3 ± 2.3 mV and 6.5 ± 1.0 mV, respectively, because the amine group of GC is protonated by hydrogen ions under acidic conditions (Figure 3b). The change of surface charge forces particles to interact with each other, disrupting the structure of the particles and making them unstable [29]. Protonation of the main lipid can be another factor that destabilizes the particles. Cholesteryl hemisuccinate (CHEMS) is a liposomal component frequently used in pH-sensitive drug release strategies [30,31,32]. These results are considered important because they relate directly to the colloidal stability of the particles when they are circulated in the blood (Appendix A).

### 2.3. Drug Release Profiles of GC-DOX/RAPA ω-Liposomes

The drug release profiles of RAPA and DOX are shown in Figure 3c. Under physiological conditions at pH 7.4, ~50% of the encapsulated RAPA was slowly released from liposomes over 24 h, however less than 8 h was sufficient at pH 4. After 24 h of incubation at pH 4, RAPA was released up to ~80%. The release of entrapped RAPA from GC-DOX/RAPA ω-liposomes was influenced by the stability of the liposomes. The hydrophobic drug RAPA is primarily located in the lipid bilayer of the liposomes [33]. The presence of acid reduced the stability of the liposomes owing to the CHEMS composition in the lipid bilayer, thus releasing RAPA rapidly, whereas it played a role as a membrane stabilizer at pH 7.4. Overall, a burst release of DOX was detected over time in the same pattern as for RAPA at pH 4 (Figure 3d). In contrast to RAPA, at pH 7.4 DOX was hardly released from GC-DOX/RAPA ω-liposomes (~20% of DOX), owing to the difference in the way the drugs were incorporated [34]. DOX was strongly conjugated to GC, making it possible to release drugs in acidic environments only. Consequently, it can be inferred that both RAPA and DOX could selectively diffuse into oncological microenvironments and release drugs from endosomal compartments.

### 2.4. Generation of Multi-Drug Resistant Cells

A DOX-resistant breast cancer cell line was derived from original parent MDA-MB-231-GFP by continuous treatment with gradually increasing concentrations of DOX for approximately 6 months [35,36]. We then observed the changes in the morphological features of both cell lines—parent cells and DOX-resistant MDA-MB-231-GFP (MDA-MB-231-GFP/DOX)—using both light microscopy and confocal microscopy (Figure 4a). Overall, compared with the parent cells, MDA-MB-231-GFP/DOX cells exhibited enlargement of the nucleus and increased cell size, and we were able to observe the presence of multinucleated cells. A number of possible mechanisms are responsible for determining the shape and size of cells, for example, one report explained that PI3K-Akt-mTOR signaling is noted as the major determinant [37]. The DOX IC_50_ value for the MDA-MB-231-GFP/DOX was 0.621 µM, which is roughly 3 times higher than that of the parent cells (0.240 µM) (Figure 4b), which shows that MDA-MB-231-GFP/DOX has significant resistance to DOX.

### 2.5. In Vitro Toxicity Analysis

To maximize the chemotherapeutic effects on DOX-resistant cells, we examined the effect of RAPA in combination with DOX. The MDA-MB-231-GFP/DOX cells were treated with serial dilutions of RAPA, DOX, and a 1:1.5 fixed ratio of DOX/RAPA entrapped in the nanoparticles, for 3 days. After incubation, their IC_50_ values and combination index (CI) were measured by MTT assay. The IC_50_ value was 74.1 nM for RAPA ω-liposomes, 856.0 nM for GC-DOX, and 20.4 nM for GC-DOX/RAPA ω- liposomes (Figure 4c). The cell viability between RAPA ω-liposomes and GC-DOX/RAPA ω- liposomes was not significantly different (0.1 ~ 2 nM of drug concentration), however, the combination of RAPA and DOX generally exhibited greater inhibition of cell proliferation than either drug alone. The CI value was significantly less than 1 (0.175), which is an indication of the high synergy between RAPA and DOX (Appendix A) [38]. Although the therapeutic effect of GC-DOX/RAPA ω- liposomes was extremely high, the additional remote loading method for DOX into liposomes could have led to a higher drug loading, thus improving the therapeutic efficacy [39]. We also investigated how RAPA can make MDA-MB-231-GFP/DOX cells overcome chemo-resistance, focusing on PI3K-Akt-mTOR signaling. RAPA treatment reduced the phosphorylation of Akt and mTOR in a dose-dependent manner (Figure 4d). These results suggest that de-phosphorylation of the mTOR signaling related target proteins may alleviate the drug resistance to DOX because they are closely related to cell proliferation. 

### 2.6. Doxorubicin Uptake and Intracellular Distribution in MDA-MB-231-GFP/DOX Cells

Sequential cell images were used to determine the intracellular doxorubicin localization in MDA-MB-231-GFP/DOX cells. The cells were incubated with GC-DOX/RAPA ω-liposomes for 0 h, 30 min, 4 h, and 8 h and then investigated using confocal microscopy. As shown in Figure 5, GC-DOX/RAPA ω-liposome treated cells exhibited DOX red fluorescence in the cytoplasm, 30 min after treatment, which indicated that GC-DOX/RAPA ω-liposomes could be effectively internalized into the cancer cells. The GC-DOX/RAPA ω-liposomes exhibited much stronger fluorescence after 4 h than after 30 min, however, DOX still remained in the cytoplasm after 4 h. After 8 h, a large amount of the drug can be seen to have migrated from the cytoplasm to the nuclei. These observations suggest that GC-DOX/RAPA ω-liposomes can deliver DOX into MDA-MB-231-GFP/DOX cells and imply that drugs can be successfully released from the nanoparticles during the cellular internalization process as pH decreases.

## 3. Materials and Methods 

### 3.1. Materials

1,2-dipalmitoyl-sn-glycero-3-phosphocholine (DPPC), cholesteryl hemisuccinate (CHEMS), glycol chitosan (GC), docosahexaenoic acid (DHA), and doxorubicin (DOX) were purchased from Sigma-Aldrich (USA). Rapamycin (RAPA) was purchased from Tokyo chemical industry (Japan). 

### 3.2. Preparation of GC-DOX/RAPA ω-Liposomes

RAPA ω-liposomes were prepared using the thin lipid film hydration method. Briefly, DPPC, CHEMS, DHA, and RAPA were dissolved in chloroform/methanol (3:1, v/v) at a 2 mg/mL total lipid concentration (Appendix A). They were mixed in a round bottom flask and evaporated under reduced pressure in a 45 °C water bath to make a lipid film. After complete drying, deionized water (pH 7.4) was added to the lipid film, the mixture was kept in an ultrasonic bath for 10 min, and then the film was dispersed using an ultrasonic probe (Q125, QSonica Llc., USA) at 50% intensity for an additional 10 min to fully hydrate the lipids and control the size of the liposomes. The crude liposomal solution was then filtered through a 0.45 µm pore membrane filter. GC-DOX was prepared according to the procedure below. Triethylamine (TEA) and cis-aconitic anhydride in DMF (53 mg/mL) were added dropwise to the DOX in DMF (12 mg/mL) solution and reacted at RT for 24 h with stirring. Excess EDC (1-ethyl-3- (3-dimethyl aminopropyl) carbodiimide) and NHS (N-hydroxy-succinimide) were added and gently stirred at RT for 1 h. Additionally, GC solution in 50% methanol (28 mg/mL) was added and gently stirred at RT for 24 h. The reaction mixture was dialyzed against distilled water for 3 days and freeze-dried. For the preparation of GC-DOX/RAPA ω-liposomes, GC-DOX was dissolved in deionized water (pH 6.5) at 2 mg/mL, then added dropwise to a RAPA ω-liposome solution (2 mg/mL) and stirred for 30 min at room temperature. The precipitates were removed by centrifugation at 4,000 rpm for 10 min.

### 3.3. Physicochemical Characterization of GC-DOX/RAPA ω-Liposomes

The particle size, size distribution, PDI, and zeta-potentials were measured by a Malvern Zetasizer (Nano S; Malvern Instruments Ltd., UK). To check pH stability, the particles were incubated in pH 4.0, pH 5, pH 6.5, or pH 7.4 PBS solutions. All measurements were performed in triplicate.

To observe the structure of GC-DOX/RAPA ω-liposomes, an aliquot of the liposome solution (10 μL of 1 mg/mL lipid) was placed on a carbon-coated 400 mesh copper grid for 10 min. The solution was removed by gentle tapping, washed twice, negative-stained, and then dried. The prepared samples were observed with an electron microscope (JEOL-2100F; JEOL Ltd., Japan).

### 3.4. Evaluation of Drug Loading and Release

DOX and RAPA encapsulation efficiency (EE) and loading efficiency (LE) were evaluated using a spectrophotometer (Infinite M200 pro; TECAN Group Ltd., Switzerland). The EE and LE were detected at wavelengths of 480 nm for DOX and 280 nm for RAPA. The EE was defined as the ratio of the actual and original amounts of drug encapsulated in the nanoparticles (NPs) and the LE was defined as the ratio of the actual weight of drug loaded and weight of the NPs containing drugs. The calculation equations are as follows:[EE (%) = Actual amount of drug in NPs/Original amount of drug in NPs × 100]
[LE (%) = Weight of actual drug in NPs/Weight of NPs containing drugs × 100]

GC-DOX/RAPA ω-liposomes were put in 100 kDa Slide-A-Lyzer Mini dialysis tubes (Thermo Fisher Scientific Inc., USA). Then, the tubes containing sample were placed in 1 L of PBS solution (pH 7.4 or pH 4) with stirring at 100 rpm at 37 °C. At designated time intervals, the tube was taken out and the samples were analyzed by spectrophotometer in triplicate.

### 3.5. Induction of DOX-Resistance in MDA-MB-231 Cells

DOX-resistant variants of MDA-MB-231-GFP cells (MDA-MB-231-GFP/DOX) were derived from the original parent cell line by continuous exposure to DOX following a dose-dependent cytotoxicity test from which the IC_50_ value was obtained (Figure 4b). Initially, the parent cell line was treated with DOX for 48 h. The cells were then subcultured and allowed to recover for a further 24 h. Each drug exposure process was carried out for approximately 1 month. The cells were maintained continuously for 6 months in the presence of DOX at the following IC_50_ concentrations; 1/120, 1/90, 1/60, 1/30, 1/10, and a stabilization period without DOX.

### 3.6. In Vitro Cytotoxicity of GC-DOX/RAPA ω-Liposomes

MDA-MB-231-GFP/DOX cells were seeded in a 96-well culture plate (5 × 10^3^ cells/well) and incubated overnight in an atmosphere of 5% CO_2_ at 37 °C. Following treatment of cells with RAPA ω-liposomes, GC-DOX, or GC-DOX/RAPA ω-liposomes, cellular viability was measured using an MTT assay. Absorbance was measured at 570 nm and IC_50_ value was calculated based on cell proliferation measurements at 72 h. Cytotoxicity in MDA-MB-231-GFP cells and MDA-MB-231-GFP/DOX cells was tested using the same method to determine their IC_50_ values.

### 3.7. Western Blot Analysis

The effects of RAPA on cells were evaluated by western blot. Proteins were extracted from the cells after drug treatment in a dose-dependent manner. We tested the phosphorylation of mTOR and Akt. The protein concentration was determined using a BCA assay kit, and electrophoresis was performed in 8% polyacrylamide gel (20 μg/lane). The proteins were transferred to nitrocellulose membranes and membranes were incubated with rabbit polyclonal antibody against phospho-mTOR (289 kDa, 1:3000) and rabbit polyclonal antibody against phospho-Akt (60 kDa, 1:3000). Mouse monoclonal antibody against β-actin (45 kDa, 1:3000) was used as the internal control. The antibody-labeled proteins were visualized using enhanced chemiluminescence (ECL) solution and detected using Chemidoc (Fusion Solo; Vilber Lourmat, France).

### 3.8. Fluorescence Microscopy

To examine the internalization of GC-DOX/RAPA ω-liposomes, MDA-MB-231-GFP/DOX cells (5 × 10^4^ cells/well) were seeded in 35 mm^2^ glass-bottom plates and grown to 60–70% confluence. To establish the location of DOX, the cells were treated with GC-DOX/ RAPA ω-liposomes and incubated for 0 h, 30 min, 4 h, and 8 h. After incubation, the cells were washed three times with PBS, fixed with 4% paraformaldehyde (PFA) for 15 min, counterstained with Hoechst 33342, and observed using confocal microscopy (BZ-8000; Keyence, Japan).

## 4. Conclusions 

We designed GC-DOX/RAPA ω-liposomes with a mean diameter of less than 200 nm and a negative zeta potential, containing both DOX and RAPA in a single nanoparticle. GC-DOX/RAPA ω-liposomes exhibited a pH-sensitive drug release profile owing to their main components, CHEMS and GC. Using in vitro tests, we verified that GC-DOX/RAPA ω-liposomes could effectively deliver the drugs into cancer cells. The combinatorial treatment with DOX and RAPA synergistically suppressed tumor growth in DOX-resistant MDA-MB-231 breast cancer cells, which were developed in this study by continuous exposure to DOX. Protein expression changes induced by RAPA treatment provided evidence that the synergistic effect is strongly related to PI3K-Akt-mTOR signaling. Furthermore, GC-DOX/RAPA ω-liposomes showed potential for in vivo application as they exhibited high colloidal stability under physiological conditions. These findings provide a viable example of using marine pharmaceuticals as a therapeutic and carrier that can control the release of cargo in acidic environments, including tumor microenvironments. The marine-derived drug delivery system consisting of chitosan and omega-3 fatty acids shows potential for the treatment of drug-resistant breast cancer.

## Figures and Tables

**Figure 1 marinedrugs-17-00581-f001:**
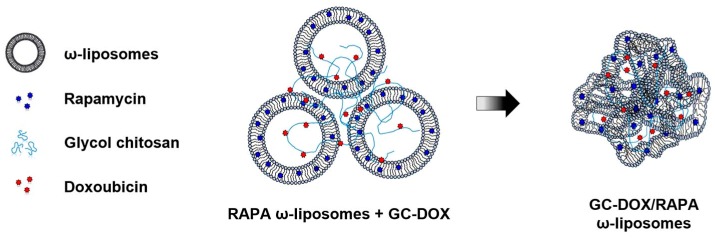
Schematic illustration of the glycol chitosan and liposome complex containing doxorubicin and rapamycin.

**Figure 2 marinedrugs-17-00581-f002:**
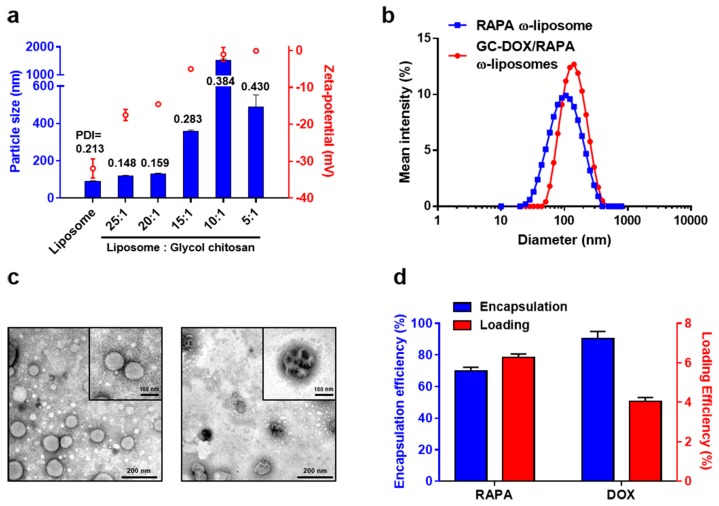
(**a**) Mean particle size (blue), zeta-potential (red), and PDI (black) values of liposomes and glycol chitosan complexes with various ratios. (**b**) Size distribution for RAPA ω-liposomes (blue) and GC-DOX/RAPA ω-liposomes (red). (**c**) Transmission electron microscope images of RAPA ω-liposomes before (left panel) and after (right panel) complexation with GC-DOX. (Scale bar: 200 nm. Zoom: 100 nm) (**d**) Encapsulation efficiency (blue) and loading efficiency (red) of RAPA and DOX in GC-DOX/RAPA ω-liposomes.

**Figure 3 marinedrugs-17-00581-f003:**
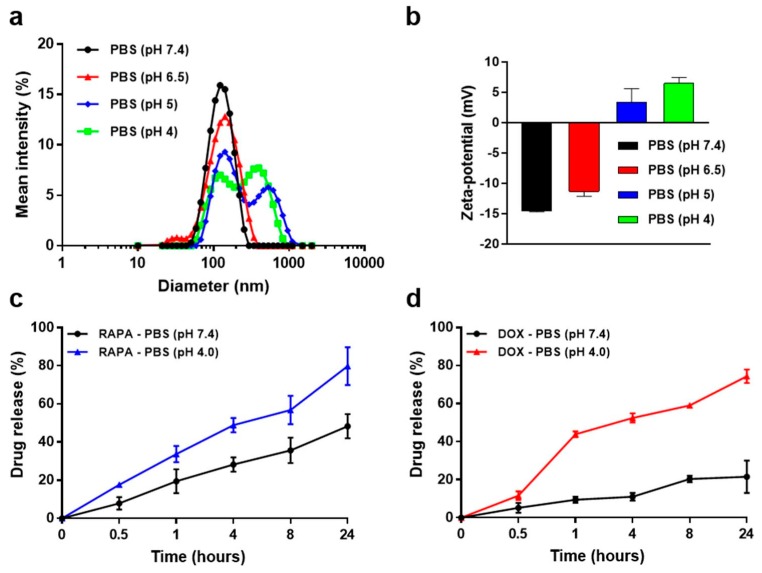
(**a**) Size distribution profiles of GC-DOX/RAPA ω-liposomes in PBS solution with different pH (each analysis was measured after 24 h incubation). (**b**) Zeta-potential change of GC-DOX/RAPA ω-liposomes in PBS solutions with different pH. (**c**) In vitro RAPA release profile of GC-DOX/RAPA ω-liposomes in PBS at pH 4.0 or 7.4 at 37 °C (n = 3/time point). (**d**) In vitro DOX release profile of GC-DOX/RAPA ω-liposomes in PBS at pH 4.0 or 7.4 at 37 °C (n = 3/time point).

**Figure 4 marinedrugs-17-00581-f004:**
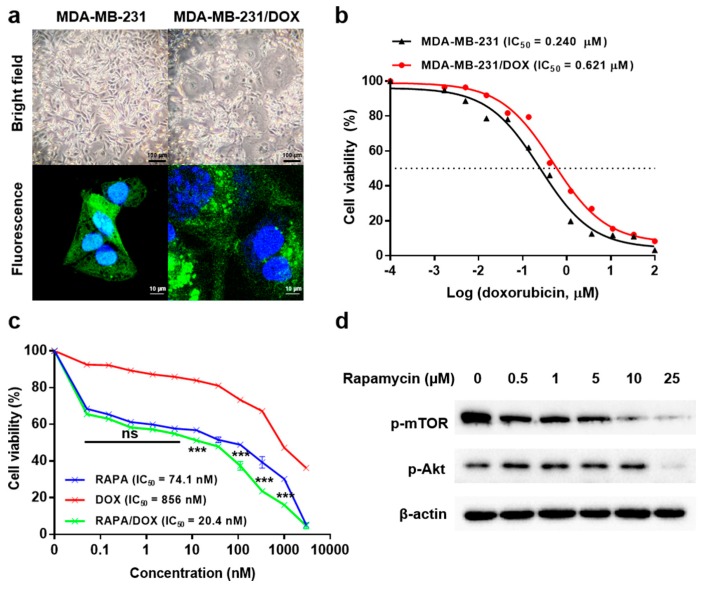
(**a**) Morphological differences between MDA-MB-231-GFP (left) and MDA-MB-231-GFP/DOX (right) cells. (**b**) Following a sustained 6-month treatment with DOX, the IC_50_ value was re-assessed for the DOX-resistant cell line relative to that of the parent cell line. (**c**) The cytotoxicity of RAPA ω-liposomes, GC-DOX, and GC-DOX/RAPA ω-liposomes at 72 h in MDA-MB-231-GFP/DOX cells. Results are presented as mean ± SEM (n = 5); *** *p* < 0.001, compared between RAPA vs RAPA/DOX (two-way ANOVA). (**d**) Expression of phosphorylated mTOR and Akt proteins in MDA-MB-231-GFP/DOX after 24 h RAPA treatment in a dose-dependent manner. The expression of each protein was assessed in RAPA ω-liposome-treated cells using β-actin as an internal control.

**Figure 5 marinedrugs-17-00581-f005:**
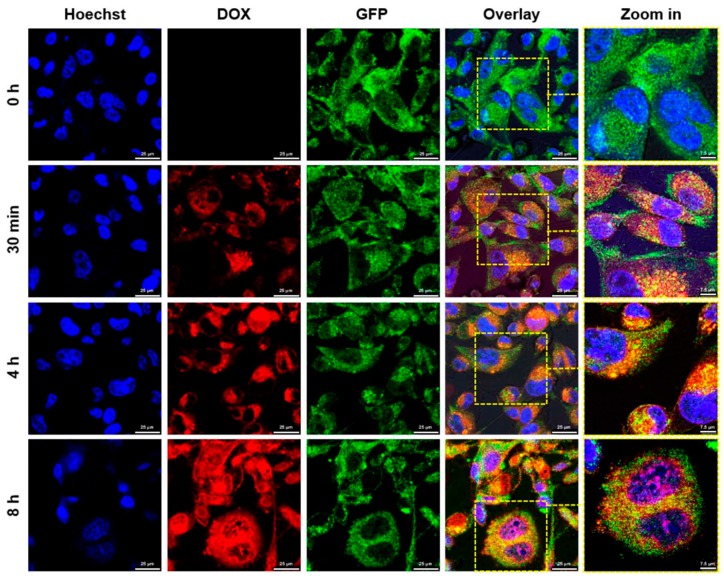
Cellular internalization of GC-DOX/RAPA ω-liposomes. Representative confocal microscopy images of MDA-MB-231-GFP/DOX cells after incubation for 0 h, 30 min, 4 h, and 8 h. MDA-MB-231 cells were stained with Hoechst 33342 (blue), doxorubicin (red), and GFP (green) (Scale bars: 25 µm, Zoom: 7.5 µm).

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
