# Peer review of "Glycol Chitosan-Docosahexaenoic Acid Liposomes for Drug Delivery: Synergistic Effect of Doxorubicin-Rapamycin in Drug-Resistant Breast Cancer"

_marinedrugs, 2019, doi:10.3390/md17100581_

Round 1

Reviewer 1 Report

I would suggest to mention in the abstract and/or introduction how this work is related to marine drugs or pharmaceuticals. Seems like it is not mentioned clearly.

What is the overall significance of this study and the results obtained from it?

Is there a specific reason why authors only selected the MDA-MD-231 cells and not others.

Can the authors demonstrate the similar results in a animal model that is relevant and applicable to the study.

Author Response

    On behalf of all authors, I appreciate the reviewers’ thoughtful comments made for our manuscript (Manuscript ID: marinedrugs-619954), entitled “Glycol chitosan-docosahexaenoic acid liposomes for drug delivery: synergistic effect of doxorubicin-rapamycin in drug-resistant breast cancer”. We amended the manuscript as the reviewer suggested.

Glycol chitosan and docosahexaenoic acid are typical marine pharmaceuticals and have been studied for cancer therapy because of their several advantages. Further discussion regarding the recent works related to glycol chitosan and docosahexaenoic acid is included in the Introduction (Page 1; 36-43). MDA-MB-231 is a well-known triple-negative breast cancer subtype and also found to exhibit the highest level of drug resistance to an array of chemotherapeutic drugs. In this study, we established a DOX-resistant MDA-MB-231 cell line and examined the effect of RAPA in combination with DOX to maximize the chemotherapeutic effect Many previous reports in regard to the examples of the animal experiment were included in Reference [6-9].

Reviewer 2 Report

The manuscript presents very interesting drug delivery system for combined release of doxorubicine and rapamycine. The study is precisely described and I recommend it for publishing after some minor changes:

- the scale bar size in Fig 2c invisible and should be improved

- the statistical analysis of cytotoxic effect of drugs loaded liposomes should be added

- The combined delivery of drugs can be enhanced by their simultaneous delivery and release in cells. The delivery system described in the manuscript is based on incorporation of rapamycin in the core of liposomes followed by electrostatic complexation of GC-DOX. Does this system provide the simultaneous intracellular drug release or drugs are released extracellularly? The difference in cell viability between cells treated with RAPA ω-liposomes and GC-DOX/RAPA ω-liposomes is not very significant in the range of 0.1 – 50 mM of drugs. Maybe the effect could be improved by introduction of DOX into the core of the liposomes during preparation of liposomes (as Dox-HCl) or in the core with rapamycine (as DOX) in liposome bilayer. Of course, apart it, the GC-DOX  can be used.

Author Response

     On behalf of all authors, I appreciate the reviewers’ thoughtful comments made for our manuscript (Manuscript ID: marinedrugs-619954), entitled “Glycol chitosan-docosahexaenoic acid liposomes for drug delivery: synergistic effect of doxorubicin-rapamycin in drug-resistant breast cancer”. We amended the manuscript as the reviewer suggested.

- As suggested, we improved the visibility of scale bars in Figure 2c and also the caption was amended.

- Statistical analyses were added in Figure 4c.

- We agree with one of the reviewer’s opinions about improving the effectiveness of the drug delivery system, especially for further animal experiments. The discussion regarding the doxorubicin loading into liposomes was included (Page 6; In vitro toxicity analysis subsection, 192-199).
